# The Impact of Essential Oil Feed Supplementation on Enteric Gas Emissions and Production Parameters from Dairy Cattle

**Angelica V. Carrazco [1], Carlyn B. Peterson [1], Yongjing Zhao [2], Yuee Pan [1], John J. McGlone [3]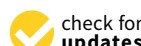, Edward J. DePeters [1] and Frank M. Mitloehner [1,*]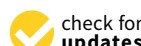**

[1]  Department of Animal Science, University of California, Davis, Davis, CA 95616-8521, USA; acarrazco@ucdavis.edu (A.V.C.); cbpeterson@ucdavis.edu (C.B.P.); yepan@ucdavis.edu (Y.P.); ejdepeters@ucdavis.edu (E.J.D.)
[2]  Air Quality Research Center, University of California, Davis, Davis, CA 95616-8521, USA; yjzhao@ucdavis.edu
[3]  Laboratory of Animal Behavior, Physiology and Welfare, Department of Animal and Food Sciences, Texas Tech University, Lubbock, TX 79409-2141, USA; john.mcglone@ttu.edu
*  Correspondence: fmmitloehner@ucdavis.edu

**Abstract:** Societal pressure to reduce enteric methane emissions from cattle continues to increase. The present study evaluated the efficacy of the commercial essential oil feed additive Agolin® Ruminant on reducing enteric gas emissions and improving milk parameters in dairy cattle. Twenty mid-lactation Holstein cows, blocked by parity and days in milk, were randomly assigned to a top dress treatment with Agolin or an un-supplemented control for a 56-day trial. Cows were group housed and individually fed twice daily. Enteric gas emissions, including methane, carbon dioxide, ammonia, and nitrous oxide, were sampled every 14 days for a 12 h period via head chambers connected to a mobile air quality laboratory. Cows supplemented with Agolin versus the control had less methane intensity (g/period/kg energy-corrected milk (ECM); $p = 0.025$). Ammonia was the most affected gas, with lower ammonia production (mg/period; $p = 0.028$), and ammonia intensity (mg/period/kg ECM; $p = 0.011$) in Agolin-fed versus control-fed cows. All cow performance variables, including dry matter intake, ECM, milk fat, milk protein, or feed efficiency were similar between treatments. Further research should evaluate how Agolin impacts ruminal flora, focusing on mechanistic impacts to fermentation.

**Keywords:** greenhouse gas; methane; essential oils; dairy cow; enteric emissions; sustainability; feed additive

## 1. Introduction

Air pollutants have strong effects on public and environmental health. Carbon dioxide ($CO_2$), methane ($CH_4$), and nitrous oxide ($N_2O$) are greenhouse gases (GHGs) that have received attention due to their contribution to climate change [1]. These GHGs, as well as ammonia ($NH_3$), can be emitted by animal agricultural processes. In an effort to regulate climate change, California passed Senate Bill 1383, mandating a 40% reduction in $CH_4$ emissions below 2013 levels by the year 2030 [2]. The primary focus of this bill lies on $CH_4$ emissions from all sources, but within the dairy sector, both $CH_4$ manure mitigation and enteric fermentation $CH_4$ can be considered. To date, the majority of research and improvements in $CH_4$ emissions have been in the area of manure management. With enteric fermentation accounting for 28% of total $CH_4$ emissions in the United States [1], reducing emissions from this source of $CH_4$ is important.

In addition to detrimental effects on the environment, $CH_4$ accounts for a 2–12% loss in gross energy intake in ruminants [3]. Dairy farmers are thus seeking ways to reduce the enteric $CH_4$ loss on their farms, and to improve cow efficiency [4]. Current strategies for reducing enteric $CH_4$ include altering the ration formulation and quality of feed [5–7], increasing amounts of dietary lipids [4], and including compounds with antimicrobial abilities such as bacteriocins or ionophores, amongst others [8,9]. Although these strategies make an impact on methanogenesis, they each come with limitations with respect to animal performance and efficiency. The use of secondary plant compounds [10], such as essential oils, has been investigated as a novel approach for reducing enteric $CH_4$ emissions [11].

Essential oils (EOs) are plant metabolites consisting of phenylpropenes and terpenes that are extracted by steam distillation or through the use of organic solvents to create a concentrated product [11,12]. These naturally occurring metabolites function to protect the plant from abiotic stress, predation, and competition, and are antimicrobial in nature [9,13]. Research suggests that EOs exhibit desirable effects with respect to rumen fermentation, thereby affecting $CH_4$ formation [14]. Individual EOs elicit varied effects on rumen fermentation and microbial persistence based on their particular modes of action [15]. Although some may also affect Gram-negative bacteria, EOs as a whole exhibit a large inhibitory effect on Gram-positive bacteria [13,16]. For instance, Chao et al. found cinnamon bark and tea tree oils inhibited both Gram-positive and -negative bacteria, fungi, and bacteriophages, whereas cedarwood and cumin oils were found to only inhibit Gram-positive bacteria. The inhibition of Gram-positive bacteria may be beneficial, because they are generally associated with the production of hydrogen ($H_2$), formate, $NH_3$, and $CH_4$ [17].

Agolin® Ruminant (AGO; Agolin SA, Bière, Switzerland), is a commercially available blend of EOs, containing coriander seed oil, eugenol, and geranyl acetate, that has been shown to reduce $CH_4$ through in vitro [18–20] and in vivo experiments [21–23]. It would be beneficial if AGO could favorably impact $CO_2$, $N_2O$, and $NH_3$, because of their importance with respect to air quality, climate, and energy loss to the animal.

Previous experiments in dairy cattle supplemented with AGO demonstrated improved milk parameters, including increased milk production [18,22] and improved milk fat and protein yield [22]. Few studies, however, have assessed the impacts of AGO on feed efficiency [18,24,25], and no previous work has addressed animal measures such as milk or serum urea nitrogen (MUN and SUN, respectively).

The present trial aimed to determine whether AGO has the potential to reduce enteric emissions of $CH_4$, $CO_2$, $N_2O$, and $NH_3$, and the potential to improve production parameters including nitrogen utilization (MUN and SUN), total milk production, and its components in mid-lactation Holstein dairy cattle. If AGO could reduce GHGs, this and related technologies could be a part of a climate change mitigation program.

## 2. Materials and Methods

### 2.1. Animals and Experimental Design

The present trial was conducted at the University of California, Davis, Dairy Teaching and Research Facility under an approved Institution for Animal Care and Use Committee protocol. The study design was a randomized complete block design with repeated measures over time. Twenty mid-lactation Holstein dairy cows were blocked according to days in milk (150 ± 43) and parity (10 primiparous and 10 multiparous), and were then randomly assigned to one of two treatments. Cows were group housed in a free-stall pen with ad libitum access to water and were milked twice daily at 6:30 a.m. and 6:30 p.m.

### 2.2. Feeding

Cows were randomly assigned to a feed gate and were individually fed using the Calan Broadbent System (American Calan, Northwood, NH, USA). Cows were fed twice daily immediately following milking at approximately 6:45 a.m. and 6:45 p.m.

The diet comprised an 89–90% dry matter (DM) total mixed ration (TMR; Table 1), with cows being fed to yield 5% refusals. Feed refusals were collected and sampled prior to each feeding at approximately 6:15 a.m. and 6:15 p.m. to determine DM content and daily DM intake (DMI). Cows were adapted to the basal diet without supplementation for 30 days prior to the start of emission sampling.

**Table 1.** Composition (%) of the basal total mixed ration (TMR) fed to cows during the 56-day trial period, as fed (89–90% dry matter (DM)).

| TMR Composition (%; As Fed) | |
| --- | --- |
| Grain Mix [1] | 41.47 |
| Alfalfa Hay | 32.25 |
| Chopped Wheat | 8.06 |
| Cottonseed, Whole | 7.68 |
| Almond Hulls | 7.68 |
| Mineral Premix | 1.15 |
| EnerGII [2] | 1.15 |
| Strata [2] | 0.32 |
| Salt | 0.23 |
| Grain Mix [1] | |
| Steam Flaked Corn | 30.75 |
| Wheat Mill Run | 21.95 |
| Dried Distillers Grains | 21.04 |
| Beet Pulp | 14.1 |
| Rolled Barley | 10.25 |
| Soybean Meal | 1.91 |

[1] Detailed composition of the grain mix and percentages of each ingredients; [2] Virtus Nutrition LLC.

The AGO and control (CON) treatments were administered as a top dress. Cows fed AGO treatment received a premix composed of cornmeal + AGO, with AGO being included at a rate of 1 g/head/day (149.5 g cornmeal + 0.5 g AGO per feeding), while CON-fed cows received 150 g of cornmeal only per feeding.

Two cows were paired in each block, with each block comprising one AGO-fed and CON-fed cow. To accommodate two cows sampled for gas emission per day in the two head chambers, cow blocks were stagger-started onto their respective treatments. Blocks were randomly assigned to a respective day 0 emission sampling day and began receiving their treatments on day 1. Treatment with AGO or CON continued for 57 days.

*2.3. Emission Sampling*

On gas emission sampling days, the two cows were each secured in their respective head chamber (HC) using neck chains similar to a tie-stall system for a 12 h gas emissions sampling period from approximately 6:45 a.m. to 6:45 p.m. Cows were subjected to three training sessions within the HC prior to the start of the experiment, in order to become habituated to the HC. Cows were sampled every 14 days, on each of their respective days 0, 14, 28, 42, and 56 on treatment. Cows had ad libitum access to feed and water and could stand up or lie down during the HC gas emission sampling period.

Each HC consisted of a head chamber manufactured with clear polycarbonate sheeting, blowers pumping air out of the hoods, and Teflon tubing to extract emission samples. The Teflon tubing was connected from the HC to a mobile agricultural air quality laboratory, which housed all of the necessary equipment [26]. Concentrations of $CH_4$, $CO_2$, $N_2O$, and $NH_3$ were analyzed using the Innova 1412 photo-acoustic multi-gas analyzer (LumaSense Technologies Inc., Ballerup, Denmark). The INNOVA 1412 analyzer had minimum detection limits of 0.4 ppm for $CH_4$, 1.5 ppm for $CO_2$, 0.03 ppm for $N_2O$, and 1.0 ppm for $NH_3$, and a maximum detection limit of 106 ppm. The continuous sampling cycle included the two HCs followed by ambient air, with each being sampled for 15 min intervals in

sequence. The HCs in use were validated by Place et al. [26], and underwent both a pre- and post-trial validation in the present trial.

## 2.4. Emission Calculations

The measured gas concentrations of the outgoing air samples from the HC for each 15 min period were truncated to remove the first five minutes and last two minutes of each sample period for the prevention of carry-over effects. The total flux of gases in mg/h were calculated according to the equation outlined in Peterson et al. with an ambient air flow rate (FL) of 2300–2500 L min$^{-1}$ [27]. The emission rate by cow for the HC (mg/h/head) was the same as the total flux of gases, as there was only one cow housed in each HC.

## 2.5. Milk Yield and Analysis

Milk yield for each cow was recorded daily at both the a.m. and p.m. milking sessions. Samples of milk were collected at consecutive a.m. and p.m. milking sessions on a weekly basis, and were sent for component analysis of milk fat, protein, and milk urea nitrogen (MUN) (Central Counties Dairy Herd Improvement Association, Atwater, CA, USA). Energy-corrected milk (ECM) was calculated according to the following formula [28]:

$$ECM = (0.327 \times milk\ kg) + (12.95 \times milk\ fat\ kg) + (7.65 \times milk\ protein\ kg). \tag{1}$$

## 2.6. Blood Sampling

Blood samples were collected from each cow following the morning milking (hour 0) and before feeding on their respective days 1, 15, 29, 43, and 57 on treatment. Animals were secured in a chute and blood samples collected into 9 mL serum separator tubes (Corvac™ Serum Separator Tubes, Covidien, Mansfield, MA, USA) from the coccygeal vein prior to returning them to the free-stall pen. Cow blood samples were immediately centrifuged, and serum was stored at −18° C. Frozen samples were transported to the UC Davis Veterinary Medical Teaching Hospital Clinical Diagnostic Laboratory Services (Davis, CA, USA) for analysis of SUN concentration.

## 2.7. Data Analysis

Gas emissions over the 12 h gas emission sampling period were summed to determine total gas production for the sampling period. Gas emissions were analyzed for intensity by calculating gas production per unit of energy-corrected milk (gas production/ECM), using ECM from the afternoon milking session which immediately followed the gas emission measurements. Gas yield was defined as the total production of the gas (i.e., summative emission measurement) per unit of DMI. Gas yield was calculated using DMI only while the cow was undergoing gas emission sampling in the HC (HC yield). Feed efficiency was calculated as kg ECM/kg daily DMI. Data were analyzed as a randomized complete block design with repeated measures, using the "nlme" and "emmeans" packages in R [29–31]. Blocks refer to each pair of parity and days in milk-matched cows. Gas emissions and production parameters were analyzed according to the following base model:

$$Y_{ijklm} = \mu + \beta_i + \beta_j + \beta_{k(j)} + \beta_l + \beta_m + (\beta_l \times \beta_m) + \varepsilon_{ijklm} \tag{2}$$

where $\mu$ = the overall mean of the response variable in question; $\beta_i$ = overall mean of day 0 for the response variable in question; $\beta_k$ = cow (random) which was nested within $\beta_j$ = block (random); $\beta_l$ = treatment; $\beta_m$ = day; $\varepsilon_{ijklm}$ = the error term. Serial correlation structures and model selection were determined based on the Akaike information criterion, Bayesian information criterion, and log-likelihood [32]. Day 0 afternoon ECM, which was used to calculate gas intensity, was unavailable for one cow; following the model selection criteria and model fit, day 0 was excluded from the gas intensity and the head chamber ECM models. The data for each of the response variables were further verified for assumptions

of normality by the Shapiro–Wilk method, with outliers removed accordingly where normality was not met.

All means are presented as least squares means (LSMs) based on "emmeans" and comparisons between treatment LSMs were completed using the "anova" function. Test day means were compared using Tukey's test pairwise comparisons using "glht" and "cld" in "multcomp" [33]. Differences were declared significant at $p < 0.05$ and a trend toward significance at $0.05 \leq p < 0.10$.

## 3. Results and Discussion

### 3.1. Effect of AGO on GHG Emissions

Methane production was found to be similar between AGO-treated versus CON-treated cows ($p > 0.05$; Table 2). Methane production differed by day ($p < 0.001$; Table S1), whereas the interaction of treatment by day was not significant. Our findings are dissimilar to those of Hart et al., who found a significant decrease in enteric $CH_4$ production when cows were supplemented with AGO [22]. Hart et al. separated the AGO-treated from the CON-treated cows in group-fed and group-treated pens, rather than individually feeding and applying the treatment to the cow. The researchers could therefore not ensure that each cow consumed the allocated 1 g/head/day of AGO in this model [22], which makes extrapolation of their findings difficult. Similarly, Castro-Montoya found that enteric $CH_4$ production tended to decrease when cows were supplemented with AGO [21]. Castro-Montoya et al. used each cow's respective day 0 as a control in their experiment; it is therefore possible that temporal changes could have affected $CH_4$ production in each cow. Klop et al. noted a brief reduction in $CH_4$ production ($p < 0.05$) in the first 14-day period after AGO supplementation began, compared with pre-treatment $CH_4$ production. Although DMI was unaffected by AGO supplementation, $CH_4$ production increased and was no longer different from pre-treatment $CH_4$ production by the third 14-day period in Klop et al. [23].

In general, a strong positive correlation has been found between $CH_4$ production and individual animal DMI [3,34]. Cows that consume higher levels of DM have more substrate available for fermentation and more hydrogen available for methanogenesis, and are therefore generally associated with higher daily $CH_4$ emissions [35,36]. Gas yield (gas emissions/DMI) is therefore an important outcome to measure [36]. In the present trial, AGO- versus CON-fed cows showed similar HC yields for $CH_4$, $CO_2$, $N_2O$, and $NH_3$ ($p > 0.05$; Table 2). Klop et al. found a reduction in $CH_4$ yield in AGO-supplemented cows when comparing the pre-treatment period to the first period (periods were 14 days in length); however, the difference was no longer present when comparing the pre-treatment period to the third or the fifth period [23]. Klop et al. housed their cows in climate respiration chambers for $CH_4$ sampling for 2.5 days, taking daily DMI into consideration. Castro-Montoya et al. found that $CH_4$ yield tended to decrease in cows supplemented with AGO ($p = 0.07$) [21]. Castro-Montoya et al. considered DMI from each of three consecutive days that animals were housed in an open circuit chamber in their calculations.

In the present study, cows supplemented with AGO versus CON showed lower $CH_4$ intensity ($p = 0.025$; Table 2). The effect of day was found to be significant for $CH_4$ intensity ($p < 0.001$; Table S1), while the interaction of treatment by day was not significant ($p > 0.05$). Our findings are consistent with those of Hart et al., who found a reduction in $CH_4$ intensity in AGO- versus CON-treated cows [22]. Klop et al. similarly noted a decrease in $CH_4$ intensity when comparing the period in which cows were on AGO treatment to the cow's respective pre-treatment period [23]. Our findings are contrary to those by Castro-Montoya et al., who found no differences in $CH_4$ intensity when cows were supplemented with AGO [21], although they used actual kg milk instead of ECM. A cow could be more productive with respect to $CH_4$ intensity; however, this is diminished as herd size increases [37].

In the present trial, no differences between AGO- versus CON-treated cows were detected for $CO_2$ production, $CO_2$ HC yield, or $CO_2$ intensity ($p > 0.05$; Table 2). The effect of day was significant for $CO_2$ production ($p < 0.001$) and intensity ($p < 0.001$; Table S1), whereas the interaction of treatment by day

was not significant for any of the $CO_2$ emission measurements ($p > 0.05$). Rumen methanogens have long been regarded nutritionally discriminatory, consuming select substrates such as $CO_2$ as a source of carbon, and $H_2$, formate, and acetate as sources of hydrogen [38]. Based on this, we would expect $CO_2$ emissions to either increase or remain unchanged by AGO supplementation. Although this was not the case, our findings were consistent with those of Melgar et al., who found decreased $CO_2$ production and no changes in $CO_2$ yield when dairy cows were supplemented with 3-nitrooxypropanol (3NOP) to reduce $CH_4$ emissions [39]. Hristov et al. saw no changes in $CO_2$ production in instances where $CH_4$ production was reduced in cows supplemented with 3NOP [39,40]. The dosing level of 3NOP was found to affect the $CO_2$ emission response, with $CO_2$ increasing as dosing levels increased [41]. Further research is therefore needed to determine if the dosage level of AGO could similarly affect $CO_2$ emissions in dairy cows.

**Table 2.** Treatment least squares means (LSMs) for gas production, gas head chamber (HC) yield, and gas intensity of methane ($CH_4$), carbon dioxide ($CO_2$), nitrous oxide ($N_2O$), and ammonia ($NH_3$) from Holstein dairy cattle to which Agolin (AGO) vs. untreated control (CON) diets were supplemented (n = 10 per treatment).

| | Treatment LSM | | SEM | *p*-Value |
| --- | --- | --- | --- | --- |
| | AGO | CON | | Treatment |
| **Gas Production** | | | | |
| $CH_4$ (g/period) | 357 | 381 | 12.1 | 0.15 |
| $CO_2$ (g/period) | 9248 | 9660 | 272 | 0.39 |
| $N_2O$ (mg/period) | 1298 | 1374 | 39.3 | 0.11 |
| $NH_3$ (mg/period) | 293 | 331 | 12.1 | **0.028** |
| **Gas Head Chamber Yield [1]** | | | | |
| $CH_4$ (g/period/kg) | 24.5 | 24.1 | 0.56 | 0.62 |
| $CO_2$ (g/period/kg) | 641 | 614 | 17.1 | 0.18 |
| $N_2O$ (mg/period/kg) | 89.1 | 87.6 | 2.12 | 0.54 |
| $NH_3$ (mg/period/kg) | 20.4 | 21.4 | 0.83 | 0.10 |
| **Gas Intensity [2]** | | | | |
| $CH_4$ (g/period/kg) | 15.8 | 17.8 | 0.71 | **0.025** |
| $CO_2$ (g/period/kg) | 411 | 452 | 22.3 | 0.15 |
| $N_2O$ (mg/period/kg) | 56.8 | 64.7 | 2.65 | 0.05 |
| $NH_3$ (mg/period/kg) | 13.1 | 15.6 | 0.82 | **0.011** |

Period = 12 h gas emission sampling period; [1] gas production per period × (1/kg dry matter intake (DMI) from the sampling period while in the HC); [2] gas production per period × (1/kg energy-corrected milk from the afternoon milking session).

No differences were found between AGO- versus CON-treated cows for $N_2O$ production, $N_2O$ HC yield, or $N_2O$ intensity in the present study ($p > 0.05$; Table 2). The effect of day was significant for $N_2O$ HC yield ($p = 0.012$), but not for $N_2O$ production or intensity ($p > 0.05$; Table S1). Similar to the other parameters, the interaction of treatment by day was not significant ($p > 0.05$). Despite making a smaller contribution to overall emissions from enteric fermentation, enteric $N_2O$ production has been quantified in the literature [42,43]. However, previous research with EO supplementation in dairy cows has not quantified enteric emissions of $N_2O$.

Enteric $NH_3$ was the most impacted gaseous emission in the present trial, with $NH_3$ production ($p = 0.028$), and $NH_3$ intensity ($p = 0.011$) being lower among AGO- versus CON-treated cows. No difference was found for $NH_3$ HC yield (Table 2). The effect of day was highly significant for $NH_3$ production, $NH_3$ HC yield, and $NH_3$ intensity ($p < 0.001$; Table S1), and the interaction of treatment by day was not significant for any of the parameters ($p > 0.05$). These findings are consistent with those of Castillejos et al., who found that the inclusion of eugenol led to a decrease in ruminal ammonia-N concentration when investigated in a batch fermentation system [44]. Coriander seed oil was also found to reduce ruminal ammonia-N concentration when compared to control- and salinomycin-treated cows [45]. The decrease in ammonia could be the result of the sensitivity of hyper-$NH_3$-producing

bacteria to EOs [11]. Working with another commercial EO, McIntosh et al. found that EOs may specifically affect the deamination of amino acids, which is the final step in protein catabolism [46]. The deamination of amino acids lead to more $NH_3$ being produced than can generally be consumed by ruminal microorganisms, resulting in nutritional losses [47]. Although $NH_3$ was measured within the ruminal fluid content for many of these studies, previous literature noted that $NH_3$ gas can form and be eructated from the rumen [48]. The reduction of $NH_3$ gas in the present trial may therefore be due to more nitrogen being retained by the animals, resulting in less nutritional loss.

Essential oils have demonstrated diverse mechanisms of action, which are used to interact with ruminal microorganisms. For example, some EOs interact with the external membranes of bacterial cells, which leads to conformational changes and the loss of stability of the cell membrane [14]. Other EOs act on microorganisms by coagulating the material within the cytoplasm of the cell [49]. The specific mechanism of action of AGO remains unclear. Further research should assess how the blend of EO within AGO individually and collectively interacts with and affects ruminal microorganisms.

### 3.2. Effect of AGO on Production Parameters

In the present study, daily DMI and head chamber DMI were similar between AGO- versus CON-treated cows ($p > 0.05$; Table 3). The effect of day on DMI was significant ($p = 0.003$; Table S2), whereas the interaction of treatment by day was not significant. Although Hart et al. found that AGO increased DMI [22], our present findings are consistent with those of both Elcoso et al. and Guasch et al., who saw no differences in DMI between treatment groups [18,25].

**Table 3.** Treatment least squares means (LSMs) for feed efficiency, daily dry matter intake (DMI), head chamber DMI, head chamber energy-corrected milk (ECM), ECM, milk fat, milk protein, milk urea nitrogen (MUN), and serum urea nitrogen (SUN) from Holstein dairy cattle fed Agolin (AGO) vs. untreated control (CON) (n = 10 per treatment).

| | Treatment LSM | | SEM | *p*-Value |
| --- | --- | --- | --- | --- |
| | AGO | CON | | Treatment |
| Feed Efficiency [1] | 1.57 | 1.63 | 0.03 | 0.28 |
| DMI (kg) [2] | 26.4 | 26.2 | 0.30 | 0.60 |
| Head Chamber DMI (kg) | 14.8 | 15.8 | 0.42 | 0.14 |
| Head Chamber ECM (kg) [3] | 22.9 | 22.0 | 1.20 | 0.49 |
| ECM (kg) | 41.1 | 42.1 | 0.98 | 0.47 |
| Milk Fat (kg) | 1.65 | 1.69 | 0.05 | 0.56 |
| Milk Protein (kg) | 1.11 | 1.13 | 0.02 | 0.60 |
| MUN (mg/dL) | 9.67 | 9.68 | 0.27 | 0.97 |
| SUN (mg/dL) [4] | 12.2 | 11.6 | 0.37 | 0.36 |

[1] kg ECM/kg daily DMI; [2] excludes DMI from when the cow was secured in the head chamber; [3] ECM from the afternoon milking session, immediately following the emission sampling period; [4] samples were collected following morning milking session (hour 0).

In the present trial, all production parameters, such as ECM, head chamber ECM, milk fat, and milk protein, were similar between treatments ($p > 0.05$; Table 3). For each of these production parameters, the effect of day was significant ($p < 0.05$), while the interaction of treatment by day was not ($p > 0.05$). Although they both focused on actual milk yield instead of ECM, our present findings are consistent with those of Castro-Montoya et al. and Santos et al., who found no differences in milk yield with AGO supplementation [21,50]. Effects of increased milk fat (kg/d) [22,50], and protein yield (kg/d) [22] have been found in previous AGO supplementation experiments; however, Castro-Montoya et al. and Elcoso et al. found no differences with respect to milk fat or protein, which is in agreement with our present findings [18,21]. In the case of Santos et al., it should be noted that the AGO treatment was applied to the pen and not the cow and the increase in milk fat yield with EO was just 0.03 kg/cow [50]. A meta-analysis conducted by Belanche et al. showed that supplementation with 1 g/head/day of

AGO to dairy cattle improved ECM (referred to as FPCM) (response ratio = 1.031; $p < 0.001$) across the 20 studies that had addressed this parameter [24]. However, it is important to note that in addition to the published literature, the meta-analysis also incorporated unpublished experiments and information from on-farm trials.

In our trial, feed efficiency was similar between AGO- and CON-treated cows (Table 3). For feed efficiency, the effect of day was found to be highly significant ($p < 0.001$; Table S2), while the interaction of treatment by day was not significant ($p > 0.05$). Elcoso et al. and Guasch et al. saw increased feed efficiency in AGO- versus CON-treated cows, which is not consistent with our present findings [18,25]. The meta-analysis conducted by Belanche et al. showed an overall improvement in feed efficiency (response ratio = 1.030; $p = 0.002$) across 16 trials, when dairy cows were supplemented with 1 g/head/day of AGO [24]. This improvement in feed efficiency appears fairly common across various EO supplements. Another commercially available blend of EO, containing eugenol, cinnamaldehyde, and capsicum, was also found to improve feed efficiency in lactating Holstein cows [51]. Supplementing cows with an EO blend containing eugenol, thymol, and m-cresol and 10 other volatile compounds, Joch et al. noted a trend towards improved feed efficiency [52]. Braun et al. also found an increase in feed efficiency when supplementing Holstein dairy cows with a commercial blend of menthol, eugenol, and anethol [53].

In the present trial, MUN was similar between dietary treatments ($p > 0.05$; Table 3). The effect of day was highly significant for MUN ($p < 0.001$; Table S2), whereas the interaction of treatment by day was not significant ($p > 0.05$). Previous studies reported varying results when supplementing cows with EOs. Benchaar et al. similarly found no differences with respect to MUN when cows were supplemented with eugenol [54], which was also confirmed by Joch et al. [52]. However, in a series of experiments where cows were supplemented with Xtract 6965 (consisting of eugenol and cinnamaldehyde) at the same dosage levels, Tekippe et al. showed increased MUN concentrations when the supplement was mixed into the mineral premix ($p < 0.001$), but not when the supplement was administered as a top dress ($p = 0.50$) [55]. Dairy cow MUN may therefore differ based on the method in which the EO is supplemented.

In the present experiment, SUN concentrations were also found to be unaffected in AGO- versus CON-treated cows (Table 3). Test day was found to be highly significant for SUN ($p < 0.001$), and the interaction of treatment by day was not significant ($p > 0.05$). Experiments conducted on Holstein dairy heifers supplemented with cinnamaldehyde demonstrated no difference in plasma urea nitrogen (PUN) between EO- versus control-supplemented cows [56]. Supplementation of eugenol and cinnamaldehyde to multiparous Holstein dairy cows resulted in inconclusive results, with significantly higher PUN concentrations when the EO was mixed into the premix ($p < 0.001$) and significantly lower PUN when the EO was applied as a top dress ($p = 0.03$) [55].

## 4. Conclusions

Societal pressure and legislation have resulted in a need for California's dairy industry to reduce GHG emissions. Our present findings suggest that supplementing lactating dairy cow rations with 1 g/head/day of AGO may be part of an effective enteric $CH_4$ intensity mitigation strategy. Agolin also demonstrated a potential for reducing nitrogen-based gas emissions in mid-lactation dairy cattle, although additional research is needed to elucidate AGO's impact on nitrogen utilization. In order to form a more comprehensive understanding of the benefits of supplementation, future research should assess AGO's impact on ruminal microorganisms, and determine the EO blend's specific mode of action.

**Supplementary Materials:** The following are available online at http://www.mdpi.com/2071-1050/12/24/10347/s1, Table S1. Test Day Least Square Means (LSM) for production, head chamber yield, gas dry matter intake with a correction factor (cDMI) yield, and intensity of methane ($CH_4$), carbon dioxide ($CO_2$), nitrous oxide ($N_2O$), and ammonia ($NH_3$) from all Holstein dairy cattle enrolled in the trial (n = 20). Table S2. Test Day Least Square Means (LSM) for feed efficiency, daily dry matter intake (DMI), energy corrected milk (ECM), Milk fat, milk protein, milk urea nitrogen (MUN), and serum urea nitrogen (SUN) from all Holstein dairy cattle enrolled in the trial (n = 20).

**Author Contributions:** Conceptualization and methodology, A.V.C., C.B.P., and F.M.M.; execution, data collection and curation, A.V.C., Y.Z., and Y.P.; formal analysis, A.V.C. and J.J.M.; writing—original draft preparation, A.V.C.; writing—review and editing, A.V.C., C.B.P, Y.Z., Y.P., E.J.D., J.J.M., and F.M.M.; supervision and project administration, A.V.C., Y.Z., and F.M.M.; funding acquisition, F.M.M.; All authors have read and agreed to the published version of the manuscript.

**Funding:** This study was funded by Agolin (Agolin SA, Bière, Switzerland) and by Feedworks USA Ltd. (Ohio, USA).

**Acknowledgments:** The authors would like to thank Hannah Bill (H.B.) for her contribution and support throughout the planning and execution of this study. We would like to thank the graduate students and interns in the Mitloehner lab for their hard work. Lastly, we would like to acknowledge Doug Gisi and the dairy staff for their support and guidance.

**Conflicts of Interest:** The sponsor played no role in the execution and interpretation of the data and preparation of the present manuscript. The authors declare no conflict of interest.

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
