# Peer review of "The Impact of Essential Oil Feed Supplementation on Enteric Gas Emissions and Production Parameters from Dairy Cattle"

_sustainability, doi:10.3390/su122410347_

Round 1

Reviewer 1 Report

In the present study the authors have evaluated the efficacy of the commercial essential oil feed additive Agolin Ruminant on reducing enteric gas emissions and improving milk parameters in dairy cattle. The study lacks novelty as the findings of this study are similar to previously published studies. The authors should have highlighted the importance of the present study in abstract as well as in conclusion section. No unusual results are being reported, so in my opinion the study is not fit for publication in Sustainability.

Author Response

In the present study the authors have evaluated the efficacy of the commercial essential oil feed additive Agolin Ruminant on reducing enteric gas emissions and improving milk parameters in dairy cattle. The study lacks novelty as the findings of this study are similar to previously published studies. The authors should have highlighted the importance of the present study in abstract as well as in conclusion section. No unusual results are being reported, so in my opinion the study is not fit for publication in Sustainability.

With Agolin being a commercially available additive, extensive research is needed to address its efficacy around production and environmental impacts. Additionally, current published literature that addresses the additives impacts on GHG focused on methane (CH4) and not at other climate pollutants. Lastly, this is the first experiment to compare individually fed control vs Agolin supplemented cows using a chamber-based approach. In summary, we respectively disagree with the reviewer and find our manuscript to make a novel contribution in an area of high public policy relevance.

Reviewer 2 Report

The impact of essential oil feed supplementation on enteric gas emissions and production parameters from dairy cattle

This manuscript details the use of Agolin®, an essential oil feed additive, that is commercially available. It tests the impact of Agolin on feed efficiency as well as other variable such as milk yield, milk fat, milk protein and milk and serum urea nitrogen. Overall, this is a well-executed and well written study. The topic is not particularly novel, excluding the factors around feed efficiency and MUN + SUN, but assessments of these commercially available products, and subsequent publication of the results is important to add to the overall body of knowledge in this field.

A (very short) list of more detailed comments are below.

Materials and Methods

Line 108 Is this the first mention of the CON group – assuming this is control but would be useful to have it detailed.

Line 119 Were the cows habituated to the HC prior to use or is it a common event for them?

Line 177 Capatilize the ‘t’ in Test

Results and Discussion

Line 185 Consider editing to clarify meaning.  Also remove the ‘-‘ if it’s not needed

Line 296 I note that the Feed Efficiency calculation is noted here. While it is obviously a simple equation, is it mentioned in the analysis material and methods (analysis sections)?

Author Response

The impact of essential oil feed supplementation on enteric gas emissions and production parameters from dairy cattle

This manuscript details the use of Agolin®, an essential oil feed additive, that is commercially available. It tests the impact of Agolin on feed efficiency as well as other variable such as milk yield, milk fat, milk protein and milk and serum urea nitrogen. Overall, this is a well-executed and well written study. The topic is not particularly novel, excluding the factors around feed efficiency and MUN + SUN, but assessments of these commercially available products, and subsequent publication of the results is important to add to the overall body of knowledge in this field.

A (very short) list of more detailed comments are below.

Materials and Methods

Line 108 Is this the first mention of the CON group – assuming this is control but would be useful to have it detailed.

Line 107 (previously line 108): We have added that this is referring to control cows. Now reads “The AGO and control (CON)…”

Line 119 Were the cows habituated to the HC prior to use or is it a common event for them?

Lines 119-120: We have added the sentence “Cows were subjected to three training sessions within the HC prior to the start of the experiment, in order to become habituated to the HC.”

Line 177 Capatilize the ‘t’ in Test

The T in “test” has been capitalized

Results and Discussion

Line 185 Consider editing to clarify meaning.  Also remove the ‘-‘ if it’s not needed

We have rephrased this to read as follows: “Our findings are dissimilar to Hart et al. who found a significant decrease in enteric CH4 production when cows were supplemented with AGO [22]. Hart et al. separated the AGO from the CON treated in group-fed and group-treated pens, rather than individually feeding and applying the treatment to the cow. The researchers could therefore not ensure that each cow consumed the allocated 1g/head/day of AGO in this model [22], which makes extrapolation of their findings difficult. Similarly, Castro-Montoya found that enteric CH4 production tended to decrease when cows were supplemented with AGO [21]. Castro-Montoya et al. used each cow’s respective day 0 as a control in their experiment; it is therefore possible that temporal changes could have affected CH4 production in each cow.”

Line 296 I note that the Feed Efficiency calculation is noted here. While it is obviously a simple equation, is it mentioned in the analysis material and methods (analysis sections)?

We have added the sentence “Feed efficiency was calculated as kg ECM/kg daily DMI." (to lines 166-167)

Reviewer 3 Report

Dear Authors,

I think it is a very interesting paper, which leads with an interesting topic nowadays. I suggest however to present the structure of the paper in the introduction. Also I suggest to add/clarify in the paper, the limitations and implications for theory and practise.

Also correct some minor errors. Pay attention to the reference of some acronyms (for example the reference to esencial oils in line 54).

Author Response

Dear Authors,

I think it is a very interesting paper, which leads with an interesting topic nowadays. I suggest however to present the structure of the paper in the introduction. Also I suggest to add/clarify in the paper, the limitations and implications for theory and practise.

We appreciate the comment and believe it is addressed in the conclusions section.

Also correct some minor errors. Pay attention to the reference of some acronyms (for example the reference to essential oils in line 54).

We have removed “(EO)” in line 52, added “(EOs)” to line 54. This now places the acronym just before its first use.

Reviewer 4 Report

 The researchers measured production parameters and used the head chamber to measure enteric gas (CH4, CO2, N2O and NH3) emission in 2 groups of dairy cows fed a the same basal TMR with supplementation of essential oil in one group. The measurements were carried out repeatedly during 12 hr periods at 14 day intervals. The manuscript is well written, but I have some issues with the way the experiment was conducted and some of the calculations:

1) Why did the authors use a 12 hr measurement  period but not 24 hr? It was clear from the numbers that DMI in the HC was higher for the control group than the AGO group which may be due to a delay in intake for cows on AGO treatment. This could have lead to the significantly higher total NH3 gas production (g/period) and numerically higher total gas (CH4, CO2, N2O)  production (g/period) among the control group. This trend may not have been observed if the measurements were made on a standard 24hr period. Though authors tried to give a reason for lower NH3 in the AGO group, on lines 269-270, it will be good if they presented data on head chamber DMI for the 2 treatments in the Tables. 

2) Two head chambers were used in the measurements of the gas emissions but there was no mention of how the cows were assigned to the 2 chambers and the chamber effect was not accounted for anywhere in the model used for analysis. An earlier studies (Troy et al 2013, Advances in Animal Biosciences, vol. 4 (2013), p. 551) reported a considerable variation in gas measurement using the head chamber. There is likely some variation between chambers which was not considered in the present study.

3) on lines 160-164, the authors used the equation from Robinson et al 2014, to calculate the corrected DMI. I do not understand why you would use that equation. Robinson and co-authors developed this equation in non-lactating, non-pregnant ewes. Even within the same species, extrapolating data from dry, non-pregnant animals to animals in production could be misleading as they do not have the similar energy requirements. Moreover, Robinson et al measured DMI intake for a period of 22 hrs, while in your study the duration of DMI measurement in the chamber was 12 hrs and the duration for the day 2 and day 1 before chamber measurement were 24 hrs. By scrutinising the use of the equation in your study, the term, DMI HC was a 12 hr measure while the terms DMI P and DMI 2P were 24 hr measures. Moreover, the equation of Robinson et al was based on feed eaten, not DMI. Are the units not supposed to be the same? The manner in which you used this equation would lead to underestimation of DMI which is obvious from the numbers you presented in Table 2. The reason provided by authors for using this equation is not adequate.

Line comments

24-cDMI data was not presented in the Tables. Please give the units of methane yield and methane intensity

88-how many of the cows were primiparous and multiparous

97-The diet comprised an 89-90%....

112-...with each block comprising one AGO-fed....

185- remove 'for' after trend

345-350-The blood samples were not taken on the day of gas measurements. So, we do not expect SUN to give any information on the rumen NH3 gas emissions which was measured during 12 hr period a day before blood sampling. The gap in intake observed between the 2 treatments during chamber measurements may have been closed before blood sampling. Dietary CP content has a greater influence on MUN and SUN, and this was similar for the 2 diets as the cows were fed the same diet except for the AGO treatment. So, the similar MUN and SUN is not unexpected.

Author Response

 The researchers measured production parameters and used the head chamber to measure enteric gas (CH4, CO2, N2O and NH3) emission in 2 groups of dairy cows fed a the same basal TMR with supplementation of essential oil in one group. The measurements were carried out repeatedly during 12 hr periods at 14 day intervals. The manuscript is well written, but I have some issues with the way the experiment was conducted and some of the calculations:

1) Why did the authors use a 12 hr measurement  period but not 24 hr? It was clear from the numbers that DMI in the HC was higher for the control group than the AGO group which may be due to a delay in intake for cows on AGO treatment. This could have lead to the significantly higher total NH3 gas production (g/period) and numerically higher total gas (CH4, CO2, N2O)  production (g/period) among the control group. This trend may not have been observed if the measurements were made on a standard 24hr period. Though authors tried to give a reason for lower NH3 in the AGO group, on lines 269-270, it will be good if they presented data on head chamber DMI for the 2 treatments in the Tables. 

We acknowledge that 24 hours of measurement would be ideal. However, we do not have the ability to milk the cows while in the head chamber. The cows go to the milking parlor. Milking cows in the milk parlor allows us to collect the milk for resale and it also is better for mammary health with respect to minimizing mastitis. Milking cows in the head chamber that is an environment associated with feed, feces, and urine is a risk for mammary infections. Cows with mastitis do not provide useful data for any parameter that we measured either in the head chamber or prior to the head chamber. The risk of mastitis that could result by milking cows in the head chamber was too great a risk. To address your last point in the comment, we have added data regarding the head chamber DMI (added to table 3).

2) Two head chambers were used in the measurements of the gas emissions but there was no mention of how the cows were assigned to the 2 chambers and the chamber effect was not accounted for anywhere in the model used for analysis. An earlier studies (Troy et al 2013, Advances in Animal Biosciences, vol. 4 (2013), p. 551) reported a considerable variation in gas measurement using the head chamber. There is likely some variation between chambers which was not considered in the present study.

A comprehensive validation study was conducted by Place et al. (2011) with the same two head chambers, which demonstrated that the head chambers function equally to one another. The chambers additionally were put through both a pre- and a post-trial validation to test for and confirm that they are in fact functioning equally to one another. This has been clarified in the manuscript in line 136.

3) on lines 160-164, the authors used the equation from Robinson et al 2014, to calculate the corrected DMI. I do not understand why you would use that equation. Robinson and co-authors developed this equation in non-lactating, non-pregnant ewes. Even within the same species, extrapolating data from dry, non-pregnant animals to animals in production could be misleading as they do not have the similar energy requirements. Moreover, Robinson et al measured DMI intake for a period of 22 hrs, while in your study the duration of DMI measurement in the chamber was 12 hrs and the duration for the day 2 and day 1 before chamber measurement were 24 hrs. By scrutinising the use of the equation in your study, the term, DMI HC was a 12 hr measure while the terms DMI P and DMI 2P were 24 hr measures. Moreover, the equation of Robinson et al was based on feed eaten, not DMI. Are the units not supposed to be the same? The manner in which you used this equation would lead to underestimation of DMI which is obvious from the numbers you presented in Table 2. The reason provided by authors for using this equation is not adequate.

We acknowledge the concerns of the reviewer for the use of cDMI. The most appropriate approach would be based on dairy cattle data for lactating cows. However, even that would still cause concerns because diet used in one study is not the same as the diet used in the current study. Also, the animal differences would also create uncertainty.

The reviewer identified dry matter intake (DMI) as numerically lower for Agolin vs CON cows; however, as the P-value shows, the difference was not significant.

The justification for using a correction for DMI (feed intake) is related to digestion kinetics and rumen fill. Ruminants, in particular dairy cattle, have numerous feeding bouts throughout the day. Feed intake is highest after milking when fresh feed is provided. The cows in the study were fed about 26 kg feed DM per day which was split into two feedings. The morning feeding of 13 kg would see a cow eat ~ 2 to 3 kg after milking in the first 30 to 60 minutes. A cow will typically follow this by getting a drink of water, and then will lie in a freestall to ruminate or will return to the manger to eat another 1 kg of feed. In about two hours, the cow is at the feed manger once again and will eat ~ 2 to 3 kg of feed. This happens throughout the day and night. Rumen fermentation is therefore continuous, with peaks and valleys in fermentation associated with new fermentable constituents consumed. Associated with the rumen fermentation/rumen fill is a rate of digestion (kd) for both particulate and soluble compounds and a rate of passage (kp) for both particulate and liquid compounds.

The rate of passage (kp) for particulates in lactating dairy cow is probably 4 to 6%/hour. So the feed that a cow consumes while in the head chamber is added to the remaining feed consumed from the day or two prior to entering the head chamber. That is the basis for the reason a correction is performed to account for the DM consumed the days prior to the head chamber.

We acknowledged that it is not perfect. Each cow consumes a different amount of DM daily. Rumen size (rumen fill) and thus kp and kd will vary per cow within and between dietary treatments. Even chewing time that we measure with rumination collars differ between cows on the same diet, and this action impacts rumen kp and kd.

In the current study the feed intake of cows the days immediately prior to the head chamber were similar. The rumen volume of Control and Agolin cows was likely similar when entering the head chamber, but the numerical volume seemed lower while in the head chamber as is reflected by the lower feed intake. The cDMI is trying to account for that difference. If the organic matter in the rumen available for fermentation is less, then enteric emissions should be lower per unit of feed.

We acknowledge the limitations for using cDMI. The cDMI attempts to account for potential differences in rumen fill since that rumen fill is potentially fermentable organic matter that contributes to enteric emission.

Line comments

24-cDMI data was not presented in the Tables. Please give the units of methane yield and methane intensity

This line has been corrected to read “Cows supplemented with Agolin versus the control had less methane corrected dry matter intake (cDMI) yield (p = 0.004) and methane intensity (p = 0.012).” In addition, units to each measurement as requested.

88-how many of the cows were primiparous and multiparous

Line 88 has been edited and now reads “(10 primiparous, 10 multiparous) …”

97-The diet comprised an 89-90%....

Line 97 has been edited and now reads “the diet comprised an 89-90%...”

112-...with each block comprising one AGO-fed....

This corresponding sentence has been changed and now reads “Two cows were paired in each block, with each block comprising one AGO-fed and CON-fed cow.”

185- remove 'for' after trend

The sentences have been reconfigured, and so no longer has “trend for”.

345-350-The blood samples were not taken on the day of gas measurements. So, we do not expect SUN to give any information on the rumen NHgas emissions which was measured during 12 hr period a day before blood sampling. The gap in intake observed between the 2 treatments during chamber measurements may have been closed before blood sampling. Dietary CP content has a greater influence on MUN and SUN, and this was similar for the 2 diets as the cows were fed the same diet except for the AGO treatment. So, the similar MUN and SUN is not unexpected.

The following has been omitted “Although the urea nitrogen findings do not appear to agree with the gaseous nitrogen findings, the proximal source of nitrogen in SUN and MUN in our present trial is unknown. The nitrogen in the MUN and BUN fractions could have derived from microbial protein that had been digested intestinally, rather than from ruminal NH3-nitrogen [59]. Future research should further address nitrogen utilization in dairy cows are supplemented with AGO.

Round 2

Reviewer 1 Report

The authors have made significant changes in the manuscript, it can be accepted in present form.

Author Response

Thanks for your review.

Reviewer 4 Report

The authors tried to justify the use of the equation of Robinson et al 2014, but they failed to convince me.

I still question the usefulness of that equation for this study. The parameters in the model are not on the same scale. First of all, the model of Robinson et al was developed based on feed eaten and not DM. You have DM in your study. Secondly, the term DMI HC is a 12 h measure while the terms DMI P and DMI 2P are 24 h measures. These periods are not congruent with the 22h periods of measurements in Robinson et al. Thus using the equation of Robinson is completely misleading. looking at the numbers in Table 2, the cDMI you calculated is woefully an underprediction (about 22% lower) of DMI.  I still don't understand why you would still use this equation?

Author Response

  •  
  • Thank you for your thorough comments - we understand what you mean by the difference in scale between the 22 h used by Robinson et al. and the 12 h used in our study. We have therefore omitted the correction to DMI and any reference to Robinson et al. from the manuscript.

Round 3

Reviewer 4 Report

I have no further comments.